# Bicyclomycin generates ROS and blocks cell division in *Escherichia coli*

**Anand Prakash**  [ORCID]*, **Dipak Dutta** [ORCID]

CSIR-Institute of Microbial Technology, Chandigarh, India

* anan.prbt17@gmail.com

**Data Availability Statement:** All relevent data are within the manuscript and its supporting Information files.

**Funding:** The authors didn't get any specific funding for this work.

## Abstract

The role of reactive oxygen species (ROS) in the killing exerted by antibiotics on bacteria is debated. Evidence attributes part of toxicity of many antibiotics to their ability to generate ROS by interfering with cellular metabolism, but some studies dismiss the role of ROS. Bicyclomycin (BCM) is a broad-spectrum antibiotic that is the only known compound to inhibit *E. coli* transcription terminator factor Rho with no known other cellular targets. In the present study, we addressed this question by checking whether the induction of oxidative stress could explain the increased sensitivity to Bicyclomycin in the *hns* deleted strain even in Δ*kil* background in *E. coli*. BCM evoked the generation of ROS in *E. coli* cells. BCM is known to cause the cell filamentation phenotype in *E. coli*. Performing fluorescence microscopic analysis, we show that bicyclomycin-dependent cell filamentation is associated with SOS response. RecA-GFP filaments were found to colocalize with the damaged DNA sites in the cell. Further analysis revealed that the genomic DNA was partitioned but the cell septum formation was severely affected under BCM treatment. Furthermore, we observed biofilm formation by *E. coli* after BCM treatment. We hypothesize that ROS production after BCM treatment could lead to cell filamentation in bacteria. A better understanding of the mode of toxicity of BCM will help us design better antibiotic treatment regimes for clinical practices, including combinatorial drug therapies. The cell filamentation phenotype observed after BCM treatment makes this antibiotic a promising drug for phage-antibiotic synergy (PAS) therapy.

## Introduction

Many antibiotics are known to employ ROS as a means of their lethality against bacteria, and a common ROS-dependent killing mechanism employed by antibiotics has been proposed [1–4], However, some researchers have negated the role of ROS in the antibacterial properties of antibiotics [5, 6]. To add to our knowledge of the involvement of ROS in antibiotic-mediated killing of pathogens, we investigated whether an inhibitor of bacterial transcription terminator Rho, bicyclomycin (BCM), causes oxidative stress in *Escherichia coli* cells. BCM (also known as Bicozamycin) is an antibiotic obtained from several *Streptomyces* species (*S. cinnamoneus*, *S. aizunensis* and *S. griseoflavus*) [7, 8]. It displays toxicity against numerous gram-negative

**Competing interests:** The authors have declared that no competing interests exist.

bacteria, viz. *E. coli*, *Salmonella*, *Enterobacter*, *Shigella*, *Neisseria*, etc. [9]. BCM has been used to treat traveler's diarrhea [10], as well as used as a veterinary medicine to treat fish, calves, and pigs in veterinary medicine [11–14].

BCM acts by inhibiting the transcription termination factor Rho [15], an essential protein in most Gram-negative bacteria [12, 16]. Two modes of BCM toxicity have been proposed; one is resulting from the inhibition of Rho function in *E. coli*, the activation of the *kil* gene product, and another by the double-stranded breaks (DSBs). The *kil* gene is carried by Rac prophage and codes for an inhibitor of cell division gene *ftsZ* [17–19]. *kil* is preceded by a rho-terminator; thus, the transcriptional suppression of *kil* expression by Rho is released by BCM [20]. Screening of single-gene knockout mutants revealed that the *hns* deleted strain is sensitive to BCM even in Δ*kil* background, suggesting that BCM toxicity has a mechanism other than *kil* gene mediated [21]. DSBs are generated due to increased collision of replisome with the transcription elongation complex (TEC) due to pervasive transcription after inhibition of Rho function [16, 22]. By checking intracellular ROS levels using dihydroethidium (DHE) and 2′,7′-dichlorodihydrofluorescein diacetate (H2DCFDA) probes and the promoter activity of *soxS*, *ahpC*, *katG*, and *oxyR* after BCM treatment, we probed whether BCM treatment could lead to oxidative stress in *E. coli* cells. ROS production inside the cell may result in a change in cell morphology. Cell elongation is a result of the SOS response [23]. DNA damage, the SOS response, and cell filamentation are interlinked. Mutants for the protease encoding gene *lon* were found to form filaments and were hypersensitive to ultraviolet radiation [24]. *lon* encodes for a protease that acts on many proteins, including the cell division regulator *sulA* [25]. DNA replication is halted for repair, and cell division is impaired when DNA damage occurs. This is why cells are quite elongated when there is damage to DNA [26]. We checked the morphology of *E. coli* in response to BCM using the lipophilic dye FM 4–64, which selectively stains the cell membrane.

Microorganisms produce biofilms in response to oxidative challenges in the environment. When subjected to oxidative stress *Mycobacterium avium* forms a biofilm [27]. Biofilm drives genetic diversity in response to oxidative stress [28]. We checked biofilm formation by *E. coli* after BCM treatment. Biofilm formation is dependent upon quorum sensing [28–30]. Quorum sensing in *E. coli* is mediated by autoinducer-2 (AI-2) [31]. BCM treatment also led to the expression of *rdar* (red, dry, and rough) colonies. The *rdar* morphotype is displayed by Enterobacteriaceae. It is a multicellular behavior of microorganisms resulting from the expression of curli fimbriae and the adhesive components of extracellular matrix [32].

## Results

### BCM treatment leads to arrested cell growth and ROS production in *E. coli* cells

Growth curve experiments were performed to check the effect of BCM on the growth of wild-type BW25113 cells. Cells showed impaired growth profiles when grown in the presence of BCM. The minimum inhibitory concentration (MIC) of BCM for the WT BW25113 strain is 37 μg/ml [33]. At a concentration of 25 μg/ml (0.7X MIC) of BCM, the wild-type cell showed an extended lag phase, and at 50 μg/ml (1.4X MIC) BCM concentration, plateau-type growth was observed without any log phase (Fig 1A). We suspected that apart from inhibiting the Rho function, the reduced growth may be due to increased ROS levels in the cells after BCM treatment. ROS levels in cells grown in the presence of BCM were probed using DHE and H$_2$DCFDA dyes. Many antibiotics are known to generate superoxide radicals by interfering with cellular metabolism in bacterial cells [1]. In our study, approximately 3-fold higher levels of superoxide were detected by DHE after antibiotic BCM treatment (Fig 1B). The *soxS* gene is

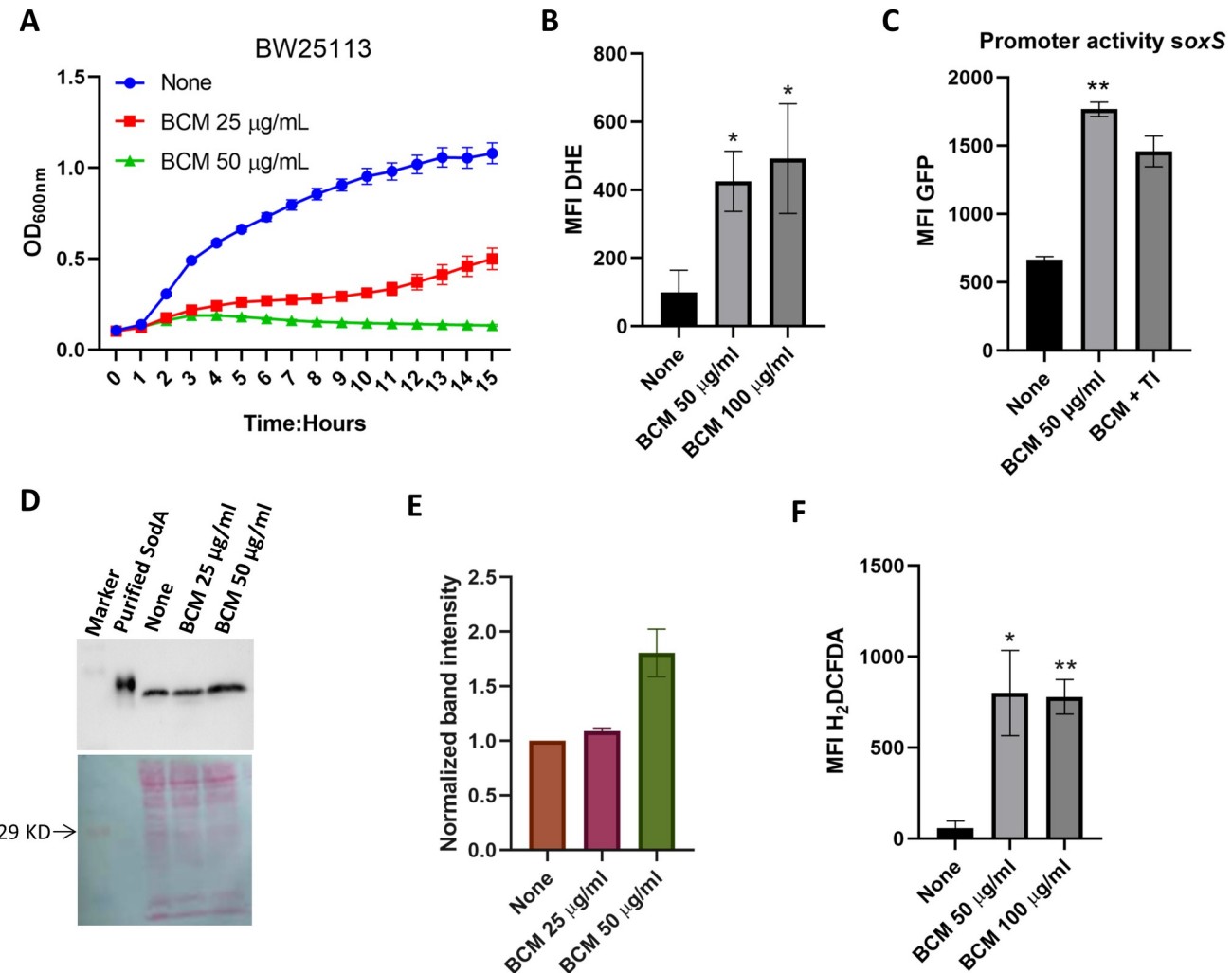

**Fig 1. WT cells exhibit retarded growth and higher ROS levels after BCM treatment. A.** WT cells show a retarded growth phenotype in the presence of BCM, Data are means ± SD (n = 3). **B.** Higher levels of superoxide were detected after BCM treatment with DHE in WT. *, $P < 0.05$. **C.** Bar graph showing higher levels of promoter activity of *soxS* after BCM treatment. **, $P < 0.01$. **D. & E.** Western blot showing 1.8-fold increase in the expression level of SodA (24 Kd) after BCM (50 μg/ml) treatment using anti-SodA antibody. Ponceau staining was performed to show equal loading of lysate in wells. Data are means ± SD (n = 2). **F.** $H_2DCFDA$ dye fluorescence increased after BCM treatment in WT. Error bars in the panels are mean ± SD from the three independent experiments. **, $P < 0.01$. *, $P < 0.05$, paired *t*-test.

induced when the intracellular levels of superoxide increase in the cells [34]. We found an increased promoter activity of *soxS* after BCM treatment. In addition, the *soxS* promoter activity was decreased when superoxide quencher tiron (TI) was added to the media (Fig 1C). In agreement with these findings, 1.8-fold increases in the SodA level were found after BCM treatment (Fig 1D and 1E).

Furthermore, the global intracellular ROS species levels were probed using H2DCFDA dye, which is pretty responsive to almost all major ROS species generated inside the cell [35]. Approximately 7-fold increases in intracellular ROS levels were detected by H2DCFDA dye after BCM treatment (Fig 1F). The promoter activities of the genes involved in the mitigation of ROS inside the cell were checked after BCM treatment using GFP-reporter constructs. We observed an increase in the promoter activity of *soxS*, *katG*, *ahpC*, and *oxyR* genes under BCM treatment (Fig 2A). The increase in the promoter activity of these genes was nullified when the

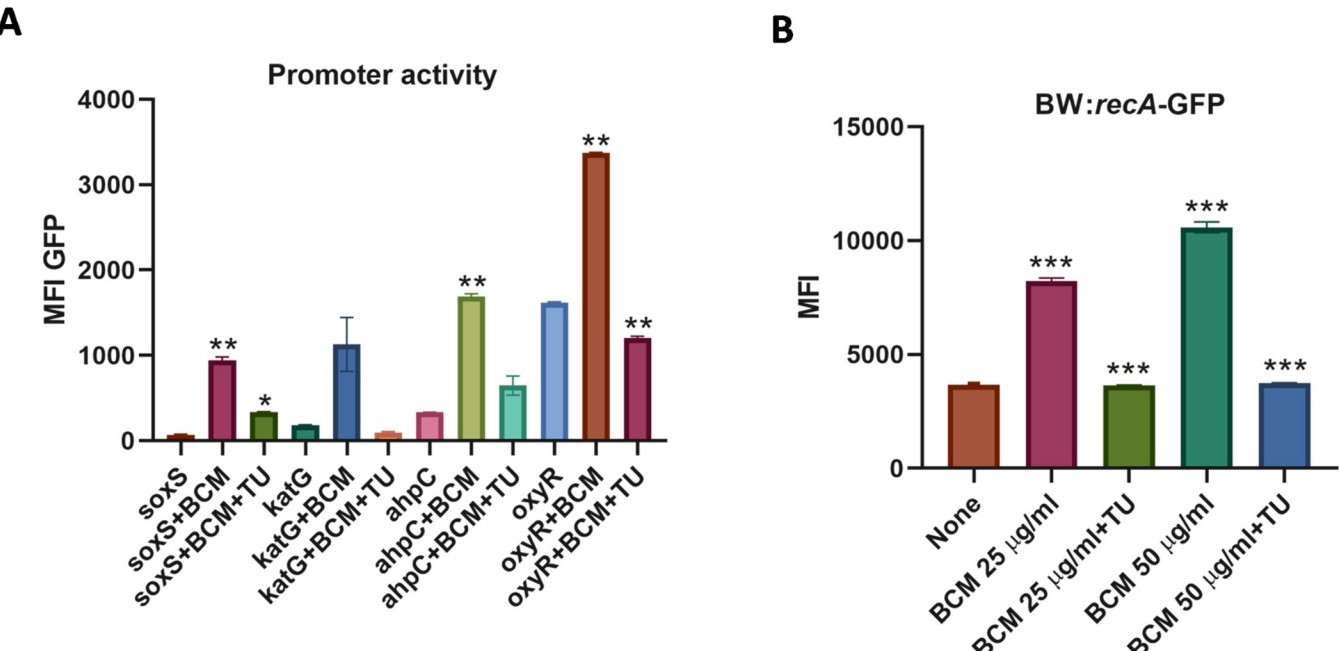

**Fig 2. BCM treatment leads to increase in promoter activity of ROS responsive genes and induction of SOS response. A.** Around 2-fold increase in the promoter activity of *soxS*, *katG*, *ahpC*, and *oxyR* genes was found after BCM treatment. **, $P < 0.01$. *, $P < 0.05$. **B.** Bar graph showing increase in RecA-GFP fluorescence after BCM treatment, supplementation of thiourea resulted in a decrease in RecA-GFP fluorescence. ***, $P < 0.001$.

hydroxyl free radical quencher thiourea was added along with BCM in the media (Fig 2A). These results suggest that exposure to BCM also leads to the generation of highly deleterious hydroxyl radicals in *E. coli* cells. Hydroxyl free radicals are most potent in damaging DNA. After quantifying the intracellular ROS production after BC treatment, we checked whether ROS generated by BCM is directly implicated in the retarded growth of *E. coli* cells. We examined the effect of BCM on the growth of Δ*sodA* Δ*katG* Δ*ahpC* triple mutant. In the absence of BCM, the triple mutant and WT grew almost identically; however, the triple mutant strain displayed increased sensitivity compared to the WT in the presence of BCM (S1 Fig in S1 File), establishing that ROS generation after BCM treatment resulted in the retarded growth profile of *E. coli* cells in the presence of the antibiotic.

## BCM treatment leads to the induction of SOS response and cell filamentation

BCM is known to cause the cell filamentation phenotype in *E. coli* [36]. In addition, BCM treatment leads to replication transcription conflicts and DNA damage [16, 22]. We checked cell septum formation, genomic DNA organization, and SOS response after BCM treatment using fluorescence microscopy. We grew WT *E. coli* containing *recA-gfp* reporter fused in the chromosome and treated with BCM. Lipophilic FM 4–64 dye was used to stain the membrane and DAPI for genomic material. Consistent with previous reports, *E. coli* cells exhibited filamentous phenotypes after treatment with BCM. Cell length increased approximately 10-fold after BCM (25 μg/ml) addition. Cell filamentation was almost abrogated when tiron was added to BCM-treated cells (Fig 3A and 3B). Inhibition of cell filament formation by tiron in the presence of BCM was in agreement with the increase in the fluorescence intensity of DHE and the increase in the promoter activity of *soxS* in the BCM-treated cells (Fig 1B and 1C). This

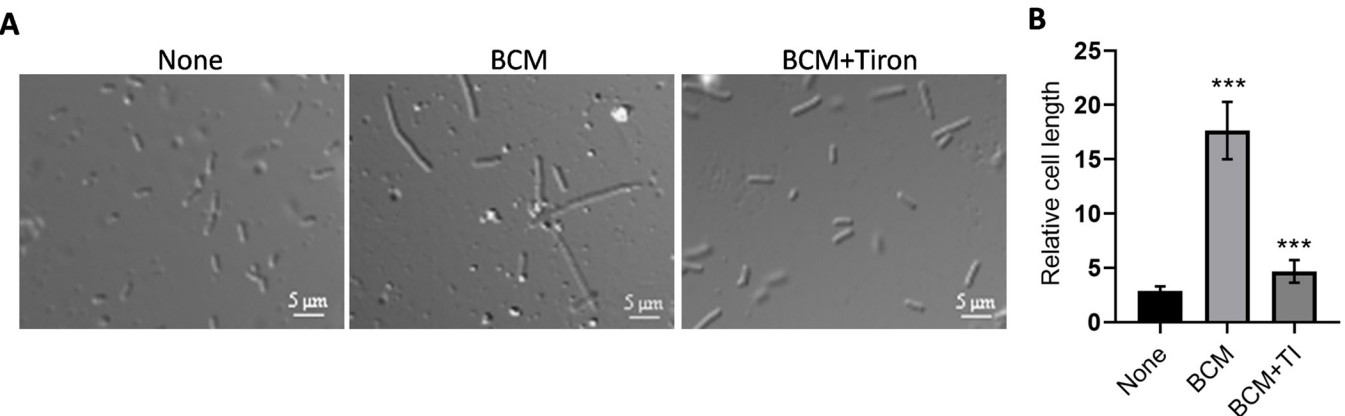

**Fig 3. Cells were highly filamented after treatment with BCM. A.** Phase-contrast microscopy image showing cell filamentation after BCM (25 µg/ml) treatment. Cell length increased approximately 10-fold in the presence of BCM. **B.** When tiron was added along with BCM, cell filamentation decreased. The calculated mean ± SD values from 30 cells were plotted. ***, $P < 0.001$.

suggests that BCM treatment leads to the superoxide formation in the cells. The effect of thiourea, another ROS quencher, was examined on cell filamentation by BCM. Thiourea treatment decreased the length of filamented cells treated with BCM (50 µg/ml) (Fig 4A and 4B). All these observations suggest that along with superoxide, hydroxyl radicals are also formed in BCM-treated cells, which lead to cell filamentation. We also checked whether ROS scavengers tiron and thiourea had any effect on the growth of *E. coli* cells. WT cells grew almost identically in the presence and absence of thiourea, and in the presence of tiron, WT cells displayed a very minor growth defect (S2 Fig in S1 File).

Incomplete septum formation was observed in the presence of BCM by the lipophilic dye FM 4–64 dye, which stains the membrane of bacteria (Fig 4C). When cells were stained with both DAPI and FM 4–64 after BCM treatment, duplication of genetic material was evident with incomplete septum formation in the cell filaments (Fig 4C). Similarly, the modulator of FtsZ cytoskeletal protein *EzrA* coding gene was 13 fold upregulated, and *ftsB*, *ftsX*, and *envC* genes involved in cell division were also moderately upregulated in microarray after BCM treatment (S2 Table in S1 File) [20]. The addition of BCM resulted in an increase of RecA-GFP fluorescence, indicating the induction of SOS response after BCM treatment (Fig 2B). RecA-GFP filaments were found to colocalize with damaged DNA sites in the cell. The colocalization of RecA-GFP foci with DAPI (Fig 4D) suggests DSBs, which is consistent with previous findings that antibiotics cause DSBs. In agreement with these results, the SOS-responsive genes, such as *dinB*, *umuD*, *sosD*, *uvrD*, and *ruvC* were upregulated to 8, 6, 10, 3, and 1.5-fold, respectively, after BCM treatment in the microarray data (S1 Table in S1 File) [20]. These observations suggest that ROS produced after BCM treatment led to the induction of SOS response in *E. coli*, due to which cell division was halted, and because of the impairment of cell division, cells become highly filamented.

## BCM treatment leads to biofilm formation

Microorganisms form a biofilm to combat environmental challenges, including oxidative stress [16, 27, 28]. Thus, after the BCM challenge, biofilm formation was checked on a microtiter plate by staining with crystal violet dye. Biofilm formation was observed after the addition of BCM. Biofilm formation increased with increasing concentration of BCM (Fig 5A and 5B). Biofilm formation is dependent on quorum sensing [29, 30]. Quorum sensing in *E. coli* is mediated by autoinducer-2 (AI-2) [31]. *luxS* synthesizes autoinducer-2 (AI-2) [37, 38]. No

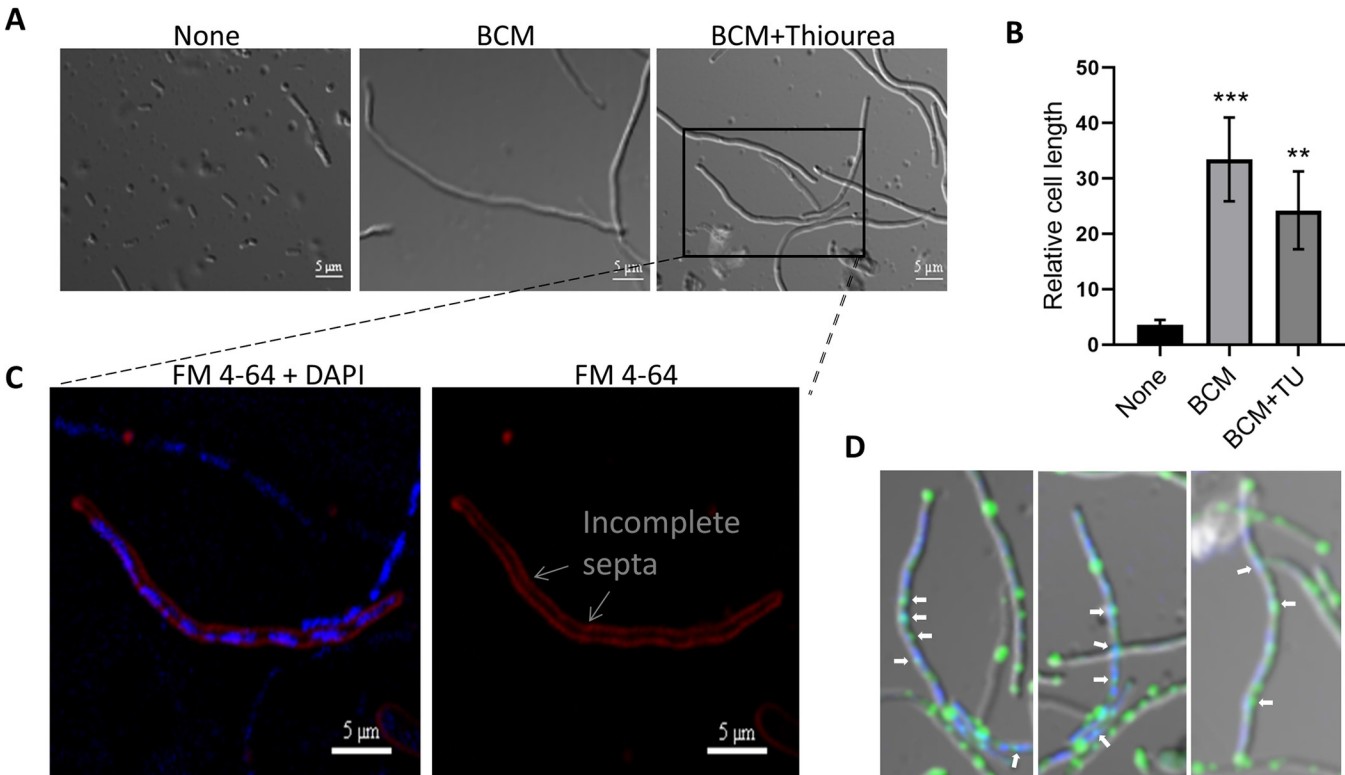

**Fig 4. BCM treatment resulted in inhibition of septum formation. A., B.** Cells became highly filamented after the addition of BCM (50 μg/ml), and after supplementation with thiourea, cell filamentation decreased. ***, $P < 0.001$. **, $P < 0.01$. The cell in the inset is zoomed to show the inhibition of septa formation and colocalization of DAPI and RecA-GFP fluorescence signals in the lower panels. **C.** Image showing partitioning of the genetic material stained by both DAPI and FM 4-64 (left panel) and abrogation of septum formation in the membrane stained by FM 4–64 dye (right panel) in the presence of BCM. **D.** RecA-GFP foci colocalized with damaged DNA stained by DAPI in presence of BCM.

biofilm was formed in the Δ*luxS* strain (Fig 5A and 5B). Biofilm formation was quantified by measuring the absorbance of the crystal violet dye at 550 nm recovered from the biofilm. The absorbance of crystal violet increased with increasing concentrations of BCM, suggesting that with a higher amount of BCM, biofilm formation by WT *E. coli* cells also increased (Fig 5B). BCM treatment also led to expression of *rdar* (red, dry, and rough) morphotype in *E. coli* colony (Fig 5C). In accordance with these results, the *luxS* and other genes involved in the biosynthesis of adhesive appendages curli (*csgA-G*) and fimbriae (*sfmA*, *sfmH*, *ydeT*, *yfcV*, *yadN*, *elfA*) were upregulated after BCM treatment in microarray in *E. coli* (S3 Table in S1 File) [20]. These results suggest that BCM treatment leads to biofilm formation in *E. coli*.

## Discussion

With ever-increasing antibacterial resistance, novel strategies and innovations are needed to tackle emerging antibiotic-tolerant pathogens. For this, a detailed understanding of the mode of action of antibiotics is crucial. A better understanding of the toxic effects of antibiotics will help us design better strategies, such as synergic drug therapy and phage-antibiotic synergy (PAS). Apart from potentiating the killing of bacteria by binding to their primary targets in cells, ROS generated by perturbation of cellular pathways by antibiotics has been uncloaked as an effective arsenal against bacteria. Norfloxacin caused superoxide-mediated oxidation of the iron-sulfur cluster and subsequent generation of hydroxyl radical breakdown through the Fenton reaction in *E. coli* [4]. Kanamycin, ampicillin, norfloxacin, and vancomycin were shown to

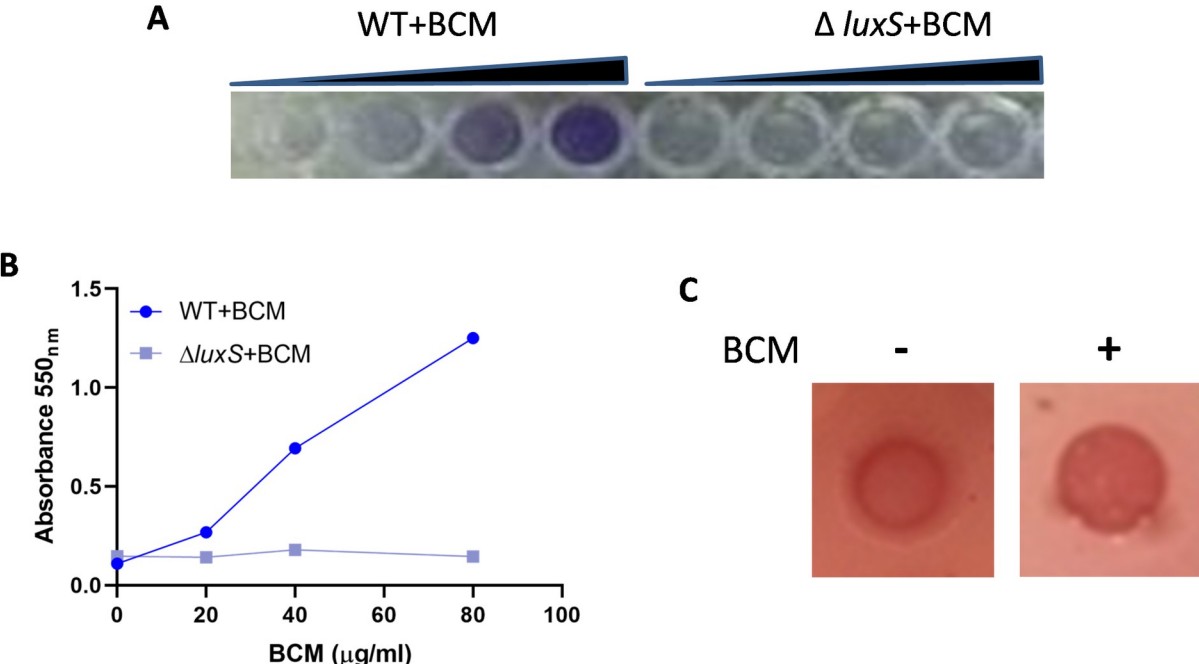

**Fig 5. *E. coli* form a biofilm in response to BCM. A.** Image showing the formation of biofilm with increasing concentrations of BCM in WT *E. coli*, but no biofilm formed in Δ*luxS* strain. **B.** Scatter plot showing the absorbance of crystal violet recovered from biofilm formed in different concentrations of BCM from WT and Δ*luxS* strain. **C.** Expression of the *rdar* morphotype was visualized on LB without salt plates supplemented with 40 μg/ml Congo Red and 20 μg/ ml Coomassie Brilliant Blue G-250 after addition of BCM.

induce hydroxyl radical formation and cell lysis in the case of ampicillin in *E. coli* cells [1]. Some studies that disapprove of the role of ROS in antibiotic-mediated killing of bacteria were fairly evaluated by considering the experimental setup and some technical aspects of the studies [3, 5, 6, 39]. BCM belongs to the 2,5dikeopiperazinies class of peptide antibiotics. Rarely observed for any antibiotic, BCM biosynthetic gene clusters are highly conserved and widespread in both Gram-negative and Gram-positive bacteria. After the discovery of BCM from *Streptomyces* species, BCM gene clusters were subsequently identified in hundreds of *Pseusomonas aeruginosa* isolates from different geographical regions, and putative BCM gene clusters have been found in seven genera of *Actinobacteria* and *Proteobacteria* (*Alphaproteobacteria*, *Betaproteobacteria*, and *Gamaproteobacteria*) [7, 40]. BCM treats diarrhea in humans, calves and pigs [10, 13, 14]. It is also economically crucial for aquaculture as it is used in the treatment of pseudotuberculosis in fish [11]. BCM is a single natural product that inhibits Rho function, along with its tested safety in mammals and its toxic effects against clinically critical pathogens like *Klebsiella pneumoniae* and *Acinetobacter baumannii*, making it an exceptionally promising antimicrobial compound [12, 33]. The hope is further enhanced by the fact that when BCM was combined with bacteriostatic concentrations of protein synthesis targeting antibiotics, it displayed a high bactericidal synergy [33]. Further, a close inspection of the structure of BCM revealed that its activity can be increased by alterations in its exomethylene group [41, 42].

Although cell filamentation after BCM treatment was reported by its discoverers as early as 1979, the reason behind this was obscure. The toxic effect of BCM is attributed to the inhibition of bacterial transcription terminator factor Rho [16, 22]. Here, we report that free radical generation is an important means of BCM toxicity, apart from the inhibition of Rho function. We found an increased ROS levels by DHE and $H_2DCFDA$ dye in *E. coli* cells after treatment

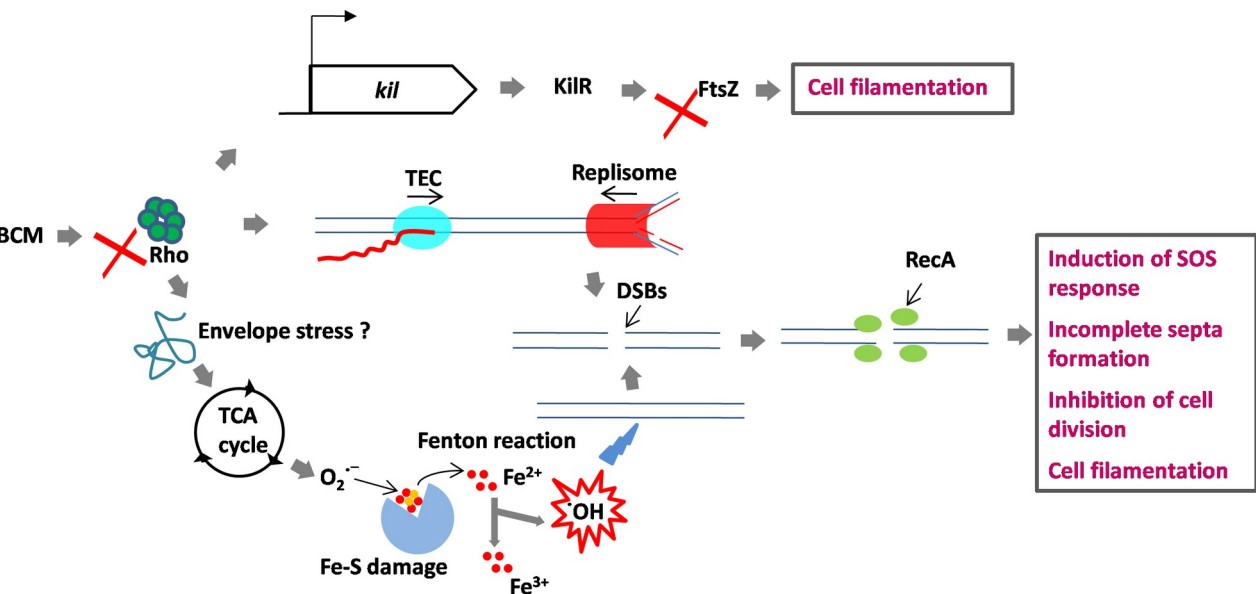

**Fig 6. H. Model for the BCM mode of action.** BCM evokes superoxide formation through the TCA cycle or envelope stress response pathway, which is converted to hydroxy radicals through the Fenton reaction. Hydroxyl radicals and the collision between TEC and replisome cause DSBs in the *E. coli* genome. At the DSBs, RecA protein is recruited and SOS response is induced. Because of the SOS response, cell division is halted and cells become filamented. *kil* expression after Rho inhibition inhibits FtsZ function, leading to cell filamentation.

with BCM. The promoter activity of *soxS*, *katG*, *ahpC, and oxyR* increased two-fold after BCM treatment. SodA levels increased 1.8-fold inside the cells after BCM treatment. Cell filamentation decreased after adding the superoxide and hydroxyl radical quenchers tiron and thiourea, respectively. These results suggest that cell filamentation observed after BCM treatment is the result of ROS production in response to BCM. Cell filaments are formed as a result of the SOS response when DNA is damaged due to environmental factors such as UV radiation or ROS [24, 26]. The cell filamentation effect of BCM on bacterial cells makes it a promising antibiotic for use in PAS in clinical practice. A previous study has demonstrated that filamentous cells are more prone to phages because of their increased surface area [43]. Overall, previous findings and our results suggest that BCM toxicity results from Rho inhibition as well as the generation of free radicals (Fig 6). The collisions between the transcription elongation complex and replisome resulting from Rho inhibition and hydroxyl radical production evoked by BCM converge to generate DSBs. The DSBs resulted in the activation of the SOS response pathway and subsequently in cell filamentation due to inhibition of cell division. Although we did not check the envelope stress after BCM treatment, it may be implicated in the generation of ROS after BCM treatment. The unregulated expression of proteins due to Rho inhibition may perturb the tricarboxylic acid cycle (TCA) cycle. Earlier, the expression of misfolded proteins has been shown to cause ROS production through the induction of envelope stress [2]. *kil* expression after Rho inhibition resulted in cell filamentation by inhibiting FtsZ function. Biofilm formation was also observed in response to BCM-induced oxidative stress. Biofilm formation was inhibited when *luxS* was deleted. Thus, biofilm formation due to BCM is *luxS*-dependent in *E. coli*. In *E. coli*, the effect of AI-2 is delivered through SdiA, which is homologous to LuxR-type transcriptional activators. AI-2-activated SdiA has been shown to regulate cell division in *E. coli* in a *ftsQAZ*-dependent manner [44]. Two modes of free radical formation by antibiotics have been deciphered. Quinolones, aminoglycosides, and β-lactams perturbed the TCA cycle, leading to low NADH levels [1]. This stimulated damage of the iron-sulfur cluster, leading to

the leaching of ferrous iron, which participated in the Fenton reaction, forming hydroxyl radicals, subsequently leading to cell death. Ribosome targeting andaminoglycoside-induced ROS production was shown to be linked with misfolded proteins in membrane and periplasmic space via envelope stress response two-component system [2]. ROS production after BCM treatment may result from metabolic stress due to enhanced expression of proteins after inhibiting Rho function. Further studies are needed to unmask the events of BCM-mediated ROS production.

## Materials and methods

### Bacterial strains and plasmids

The wild-type BW25113 (WT) strain of *E. coli* is from the KEIO collection [45]. *recA-gfp* was freshly transduced into the WT genome by P1 phage transduction to obtain the BW25113::*recA-gfp* strain [46]. The reporter plasmids pUA66_soxS, pUA66_ahpC and pUA66_katG were provided by Dr. Csaba Pal, Biological Research Centre, Hungarian Academy of Sciences [47].

### Chemicals

Dihydroethidium (DHE) and 2´, 7'-dichloro-dihydrofluorescein diacetate (H$_2$DCFDA), FM 4–64, Thiourea, Crystal violet, Congo red, 4,5-Dihydroxy-1, - 3-benzenedisulfonic acid (Tiron) and Bicyclomycin were purchased from Sigma.

### Growth conditions

The primary culture was grown overnight under shaking conditions at 37°C. The overnight grown primary culture of WT cells was inoculated into fresh LB broth at 100-fold dilution. The secondary cultures were grown for 1.5 h at 37°C till the O.D.$_{600}$ reached approximately 0.3. BCM (25 µg/ml or 50 µg/ml) was added at this point and incubated for 3 h at 37°C. The bacterial pellets were collected, and different assays were performed.

### Growth curve analyses

Saturated overnight culture of WT *E. coli* cells was diluted 100-fold in 1 ml of LB broth and grown in the presence and absence of BCM (25µg/ml or 50 µg/ml) in triplicate. The growth curves were followed for 16 h using an automated BioscreenC growth analyzer (Oy growth curves Ab Ltd.) with shaking at 37°C. The mean of the three OD at 600 nm were plotted against time to obtain the growth profile of WT cells in the presence of BCM.

### Checking ROS level and induction of SOS response by flow cytometry

The relative ROS level in WT BW25113 cells was probed using DHE and H2DCFDA dye. DHE is sensitive to superoxide, and H2DCFDA is sensitive to hydroden peroxide and hydroxyl radicals, and some other ROS species. Cells were grown for 4 h in the presence and absence of BCM (25µg/ml or 50 µg/ml) and the pellet was washed with 1X PBS. One part of the washed pellet was stained by 2 µM DHE, the other by 10 µM H$_2$DCFDA for one hour, the third was dissolved in 1X PBS. RecA-GFP fluorescence after BCM treatment was checked using a specific *E. coli* strain that harbors an SOS responsive BW25113::*recA-gfp* fusion in the genome. The data were acquired using FACS accuri (BD) with 0.1 million cells. FL1 filter was used for H2DCFDA and RecA-GFP, and FL2 for DHE. Mean fluorescence intensity (MFI) values obtained from three experiments were plotted after subtracting background fluorescence.

## Checking promoter activity of ROS-responsive genes

GFP-mut2 containing reporter plasmids, pUA66_katG, pUA66_ahpC, pUA66_oxyR, and pUA66_soxS, were transformed into the BW25113 WT strain. The transformed cells were grown with or without supplementation of BCM at 37˚C in shaking. BCM (50μg/ml) was added from starting with or without thiourea (70 mM),and tiron (40μM). The data were acquired using FACS acuri (BD) at FL1 laser for 0.1 million cells. MFI values obtained from two experiments were plotted after subtracting background fluorescence.

## Checking intracellular SodA levels after BCM treatment

WT *E. coli* culture was grown overnight, inoculated in LB medium at 1:100 dilution, and allowed to grow at 37˚C for 1.5 h under shaking conditions. BCM was added and incubated for 3 h at 37˚C. Cells were pelleted and lysed using B-PER Reagent (Thermo scientific) supplemented with 100 μg/ml lysozyme and 1 mM PMSF, followed by sonication. 35 μg of lysate was loaded on 10% SDS-PAGE after quantifying the protein concentration using Bradford assay kit (Bio-Rad). After transferring the bands to nitrocellulose membrane, Ponceau S staining was performed to check for equal loading of protein on SDS-PAGE. 5% skimmed milk was used for blocking to avoid nonspecific binding. The membrane was incubated with polyclonal rabbit primary antibodies raised against purified SodA protein. After washing 3 times with TBST, the membrane was incubated for 1 h with HRP-conjugated anti-rabbit secondary antibody. Immobilon Forte Western HRP substrate (Millipore) was used to develop the blot.

## Visualization of filament formation by confocal microscopy

WT cells harboring *recA-gfp* were grown for 1.5 h with or without 50 μg/ml of BCM. 10 μM of Tiron and 70 mM of thiourea were added wherever required. Cell pellets were washed with 1X PBS. Additionally, cells were stained with DAPI and FM 4–64 for half an hour to visualize genetic material and cell membrane, respectively. After fixing the cells with 4% formaldehyde, slides were prepared using 10 μl of sample and examined using Nikon confocal microscope using 488 nm laser for GFP, 405 nm laser for DAPI and, 560 nm for FM 4–64. Image J. software was used to measure the relative cell lengths from around 30 cells.

## Checking biofilm formation after BCM treatment

Biofilm formation after BCM treatment was checked in a microtiter plate. WT cells were seeded in 96 well plates in 200 μl LB medium with increasing concentration of BCM and incubated at 37˚C. After 24 h the media was decanted and washed with distilled water to remove any unattached cells. 125 μl of a 0.1% solution of crystal violet (CV) in water was used to stain the biofilm for 10 min. The plate was rinsed with water 3 times and blotted on tissue paper to remove any extra stain. The biofilm was photographed after drying the plate overnight. Next, biofilm formation with increasing concentration of BCM was quantified by estimating the amount of CV dye bound to it. CV was solubilized by adding 125 μl of 30% acetic acid to each well of the microtiter plate and kept for 10 min. The solubilized CV was transferred to a new 96-well plate. Absorbance was measured using BiotekPowerWave ™ XS plate Reader at 550 nm and plotted. To check the effect of BCM on colony morphology, LB without salt agar plates were prepared. Expression of the (red, dry, and rough) *rdar* morphotype was visualized on LB without salt plates supplemented with 40 μg/ml Congo Red and 20 μg/ ml Coomassie Brilliant Blue G-250.

## Statistical analyses

Experiments were performed in triplicate. Data analyses and normalization were done using Microsoft Excel, and graphs were plotted using GraphPad Prism software. Cell length measurement and densitometry of western blots were done using imageJ software. *P* values were calculated from the paired *t*- test, and the values $< 0.05$ were considered significant. Data are plotted as means ± standaed deviation (SD).

## Supporting information

**S1 File.**
(DOCX)

**S2 File.**
(XLSX)

## Acknowledgments

AP was a senior research fellow, DBT. AP is thankful to Mr. Deepak Bhatt (CSIR-IMTECH) for helping in acquisition of confocal images.

## Author Contributions

**Conceptualization:** Anand Prakash.

**Data curation:** Anand Prakash.

**Formal analysis:** Anand Prakash.

**Funding acquisition:** Dipak Dutta.

**Investigation:** Anand Prakash.

**Methodology:** Anand Prakash.

**Project administration:** Anand Prakash.

**Resources:** Dipak Dutta.

**Software:** Anand Prakash.

**Validation:** Anand Prakash.

**Visualization:** Anand Prakash.

**Writing – original draft:** Anand Prakash.

**Writing – review & editing:** Anand Prakash, Dipak Dutta.

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
