## [Decision Letter · Decision Letter 0]

28 Nov 2023

PONE-D-23-34381Bicyclomycin generates ROS and blocks cell division in Escherichia coliPLOS ONE

Dear Dr. Prakash,

Thank you for submitting your manuscript to PLOS ONE. After careful consideration, we feel that it has merit but does not fully meet PLOS ONE’s publication criteria as it currently stands. Therefore, we invite you to submit a revised version of the manuscript that addresses the points raised during the review process.

**ACADEMIC EDITOR: **

Dear author,

Revise the whole manuscript carefully in light of both reviewer's comments and submit it again.

We look forward to receiving your revised manuscript.

Kind regards,

Samiullah Khan, Ph. D

Academic Editor

PLOS ONE

Journal Requirements:

"This work received funding from intramural grants of CSIR-IMTECH (OLP-0190) and extramural grant to DD from SERB (CRG/2019/001174), Dept. of Science and Technology (DST)."

"This work received funding from intramural grants of CSIR-IMTECH (OLP-0190) and extramural grant to DD from SERB (CRG/2019/001174), Dept. of Science and Technology (DST)."

"This work received funding from intramural grants of CSIR-IMTECH (OLP-0190) and extramural grant to DD from SERB (CRG/2019/001174), Dept. of Science and Technology (DST)."

6. Please note that in order to use the direct billing option the corresponding author must be affiliated with the chosen institute. Please either amend your manuscript to change the affiliation or corresponding author, or email us at plosone@plos.org with a request to remove this option.

Additional Editor Comments:

Dear author,

Revise the whole manuscript in light of both reviewer's comments and submit again.

Reviewers' comments:

Reviewer's Responses to Questions

**Comments to the Author**

1. Is the manuscript technically sound, and do the data support the conclusions?

Reviewer #1: Yes

Reviewer #2: Yes

2. Has the statistical analysis been performed appropriately and rigorously? 

Reviewer #1: Yes

Reviewer #2: No

3. Have the authors made all data underlying the findings in their manuscript fully available?

Reviewer #1: Yes

Reviewer #2: Yes

4. Is the manuscript presented in an intelligible fashion and written in standard English?

Reviewer #1: Yes

Reviewer #2: Yes

5. Review Comments to the Author

Reviewer #1: Methods

- The “Statistical data analysis” section is missing.

Results

- Figure 1.A shows the effect of BCM on the growth of E. coli by measuring OD600 at concentrations of 25 and 50 µg/ml. This indicator reflects the bacteriostatic effect of BCM. However, in this study, the bactericidal effect of antibiotic is more important. It is therefore desirable, in this and other cases, to additionally measure the effect of BCM on the colony forming ability (CFU) of E. coli at concentrations of 25, 50 and 100 µg/ml.

- Figure 1.A shows the effects of 25 and 50 µg/ml BCM, Figure 1.B shows the effects of 50 and 100 µg/m, and Figure 1.C does not show BCM concentrations at all.

- In Supplementary (table 1-3), fluorescence values in the control (without antibiotic) should be indicated.

- In experiments with the addition of antioxidants, it should be taken into account that in the concentrations used can significantly inhibit the growth of bacteria. As is known, even a slight decrease in growth can increase the resistance of bacteria to the action of bactericidal antibiotics. As a result, separate experiments must be performed to determine the effect of antioxidants on both promoter expression and the growth and survival of bacteria treated with antioxidants alone.

- The authors write:

“…Here, we report that apart from the inhibition of Rho function, free radical generation is an important means of BCM toxicity…

…Overall, previous findings and our results suggest that BCM toxicity results from Rho inhibition as well as the generation of free radicals..”.

The authors established the generation of reactive oxygen species by BCM by several methods, but the question of whether these ROS concentrations can have a toxic effect remains open. The toxic effects of bactericidal antibiotics include effects such as reduced survival and lysis, whereas the authors showed only a bacteriostatic effect. The production of ROS in itself does not necessarily lead to a bactericidal effect. Bacteria have powerful antioxidant defenses to prevent the toxic effects of sufficiently high concentrations of ROS.

The study revealed a number of effects associated with the action of BCM, but in order to establish their connection with ROS, additional experiments are required, in particular using mutants in antioxidant systems.

Reviewer #2: Dear Editor,

This is an interesting work that deepens and provides new information on the molecular and mechanistic bases in relation to the antimicrobial activity of bicyclomycin. In this work, various experimental approaches are carried out that provide new information of interest. Knowledge of the metabolic response to antimicrobials beyond their interaction with their primary target is a current topic that will be useful for the design of new therapeutic strategies.

Below are various points and comments that I understand are necessary for a better presentation of this work.

Major points

It is necessary to clarify or justify the concentrations of BMC used. Although it is mentioned that concentrations of 25 and 50 μg/ml of BCM were used, it is important to indicate them in relation to the minimum inhibitory concentration (MIC) of this drug. I suspect that it is equivalent to 1/2x and 1x MIC but the CMI value should be specified clearly.

Results (L5): I disagree with this sentence: “We suspected that the reduced growth may be due to increased ROS levels in the cells after BCM treatment.” The main molecular mechanism of growth inhibition by BMC is already known (Rho inhibition). Please rewrite this sentence.

I have important doubts with the quality of the images provided, at least in the version that I have been able to view. Please improve the quality of the images, especially Figure 1D, 4D, 5A and 4C. Additionally, I think that Figure 1F could be incorporated into Figure 2.

An important aspect of the work is that it should be more clearly specified what type of reactive oxygen species are detected by the DHE and H2DCFDA probes. This should be described precisely in Methods. In relation to this, the Results statement "These results suggest that exposure to BCM also leads to the generation of highly deleterious hydroxyl radicals" is not clear. Why do we talk about "hydroxyl radicals" specifically if H2DCFDA is a generic ROS marker?

Please delete this sentence from Discussion; “Although, we did not check the envelope stress after BCM treatment, it may be implicated in the generation of ROS after BCM treatment.” It is excessively speculative.

Methods: In relation to this sentence: “recAgfp was freshly transduced into the WT genome by P1 phage transduction to obtain the BW25113::recA-gfp strain (45).” Please, could you indicate the precise integration site in the bacterial chromosome?

It is necessary to include a Statistical Analysis section in the Methods section where the analyzes carried out are specified.

Minor points

Abstract: BMC should be used always after its definition

Abstract: “PAS” is not defined in any part of the text.

Introduction:

L4: References (1-4) does not include information about “some researchers have negated the role of ROS…”

L6: BMC is mentioned at the fort time in the test, please define.

L8: Please change “gram-negative” to “Gram-negative”

Results:

L2: Please change “Effect of BCM” to “effect of BCM”

Figure 1C: Please, change Soxs to SoxS

Please change “TheBCM is known to cause the cell filamentation” to “The BCM is known to cause the cell filamentation”.

Please change “upregulated many-fold in microarray after BCM treatment” to “upregulated up to XX-fold in microarray after BCM treatment”.

Please change “in WT E. coli (Figure 5.B).. BCM” to “in WT E. coli (Figure 5.B). BCM”.

Discussion

Please change “ampicillin, Norfloxacin, and vancomycin” to “ampicillin, norfloxacin, and vancomycin”.

Please change “and H2DCFDA dye in Escherichia coli cells” to “and H2DCFDA dye in E. coli cells”.

Please change “deciphered. Quinones, aminoglycosides” to “deciphered. Quinolones, aminoglycosides”.

In the sentence: “MFI values obtained from three experiments”. MFI is not defined in any place.

In the sentence: “Forte Western HRP substrate (Millipore) was used to develop the”. Correct the end of the sentence.

6. PLOS authors have the option to publish the peer review history of their article (what does this mean?). If published, this will include your full peer review and any attached files.

Reviewer #1: No

Reviewer #2: **Yes: **José Manuel Rodríguez-Martínez

---

## [Author Response · Author response to Decision Letter 0]

22 Jan 2024

We thank the Editor and reviewers for the constructive criticism. Below please find point by point responses to yours and their concerns. 

Editor’s comments

The co-author of this manuscript Dr. Dipak Dutta edited this manuscript.

"This work received funding from intramural grants of CSIR-IMTECH (OLP-0190) and extramural grant to DD from SERB (CRG/2019/001174), Dept. of Science and Technology (DST)."

"This work received funding from intramural grants of CSIR-IMTECH (OLP-0190) and extramural grant to DD from SERB (CRG/2019/001174), Dept. of Science and Technology (DST)."

"This work received funding from intramural grants of CSIR-IMTECH (OLP-0190) and extramural grant to DD from SERB (CRG/2019/001174), Dept. of Science and Technology (DST)."

The funding related text has been now removed from the manuscript. I want to update the funding statement as “The authors didn’t get any specific funding for this work.”

6. Please note that in order to use the direct billing option the corresponding author must be affiliated with the chosen institute. Please either amend your manuscript to change the affiliation or corresponding author, or email us at plosone@plos.org with a request to remove this option.

I request to remove the direct billing option.

Reviewer #1: Methods

- The “Statistical data analysis” section is missing.

Statistical data analysis is now presented.

Results

- Figure 1.A shows the effect of BCM on the growth of E. coli by measuring OD600 at concentrations of 25 and 50 µg/ml. This indicator reflects the bacteriostatic effect of BCM. However, in this study, the bactericidal effect of antibiotic is more important. It is therefore desirable, in this and other cases, to additionally measure the effect of BCM on the colony forming ability (CFU) of E. coli at concentrations of 25, 50 and 100 µg/ml.

Previously CFU assays, it has been found that BCM shows very little bactericidal effect if any, even in the very high concentrations range of 200 and 400 µg/ml (Malik et al 2014). Considering the economical aspect of highly priced BCM, instead of repeating the previously done experiments, we consider that reduced growth of bacteria after BCM treatment in our growth curve experiment is in agreement with the finding of the previous study, where it was shown that BCM exerts only bacteriostatic effect. 

- Figure 1.A shows the effects of 25 and 50 µg/ml BCM, Figure 1.B shows the effects of 50 and 100 µg/m, and Figure 1.C does not show BCM concentrations at all.

Concentration of BCM is now shown in Figure 1.C.

- In Supplementary (table 1-3), fluorescence values in the control (without antibiotic) should be indicated.

The table shows fold change in the values of fluorescence after antibiotic treatment. The fold change was generated by dividing the florescence value got after antibiotic treatment with the fluorescence values without antibiotic treatment.

- In experiments with the addition of antioxidants, it should be taken into account that in the concentrations used can significantly inhibit the growth of bacteria. As is known, even a slight decrease in growth can increase the resistance of bacteria to the action of bactericidal antibiotics. As a result, separate experiments must be performed to determine the effect of antioxidants on both promoter expression and the growth and survival of bacteria treated with antioxidants alone.

Thanks for suggestion. Growth curve experiments have been now performed in the presence of antioxidants alone and put in supplementary as supplementary Figure 1 B. Addition of tiron has no effect on the growth of WT, growth was exactly same as without tiron. While growth in presence of thiourea mimicked same growth profile with very negligible shift. If we consider this shift responsible for increased resistance of the bacteria to the action of antibiotic then this can’t explain the increase in the resistance to the antibiotic after addition of tiron, as growth profile of the WT after addition of tiron just superimposed on the growth profile without addition of tiron. 

Figure 1.B. Checking effect of ROS scavengers, tiron and thiourea on the growth of WT BW25113. WT cells grew almost identically without and with supplementation of thiourea. When tiron was added to the media, WT cells showed very little, if any, slow growth.

- The authors write:

“…Here, we report that apart from the inhibition of Rho function, free radical generation is an important means of BCM toxicity…

…Overall, previous findings and our results suggest that BCM toxicity results from Rho inhibition as well as the generation of free radicals..”.

The authors established the generation of reactive oxygen species by BCM by several methods, but the question of whether these ROS concentrations can have a toxic effect remains open. The toxic effects of bactericidal antibiotics include effects such as reduced survival and lysis, whereas the authors showed only a bacteriostatic effect. The production of ROS in itself does not necessarily lead to a bactericidal effect. Bacteria have powerful antioxidant defenses to prevent the toxic effects of sufficiently high concentrations of ROS.

The study revealed a number of effects associated with the action of BCM, but in order to establish their connection with ROS, additional experiments are required, in particular using mutants in antioxidant systems.

Thanks for suggestions. We have now checked the effect of BCM on the growth of ΔsodAΔKatGΔahpC triple mutant. Without addition of BCM the triple mutant showed growth identical to the WT, but after addition of BCM the triple mutant displayed reduced growth in comparison to WT (Supplementary Figure 1A). 

Figure 1 : A. ΔsodA ΔkatG ΔahpC triple mutant (Δ3) is more sensitive to BCM. WT and Δ3 strains displayed almost identical growth profiles in the absence of BCM. However, Δ3 strain showed more reduced growth in the presence of BCM compared to WT.

Reviewer #2: Dear Editor,

This is an interesting work that deepens and provides new information on the molecular and mechanistic bases in relation to the antimicrobial activity of bicyclomycin. In this work, various experimental approaches are carried out that provide new information of interest. Knowledge of the metabolic response to antimicrobials beyond their interaction with their primary target is a current topic that will be useful for the design of new therapeutic strategies.

Below are various points and comments that I understand are necessary for a better presentation of this work.

Major points

It is necessary to clarify or justify the concentrations of BMC used. Although it is mentioned that concentrations of 25 and 50 μg/ml of BCM were used, it is important to indicate them in relation to the minimum inhibitory concentration (MIC) of this drug. I suspect that it is equivalent to 1/2x and 1x MIC but the CMI value should be specified clearly.

We agree that the concentration of BCM should have been used in respect to the MIC. Inspired by a study (Cardinale et al 2008), for which they used 10 μg/ml, 25 μg/ml and 100 μg/ml of BCM, we were more absorbed in deciphering the role of Rho in various cellular metabolism like heat shock response, manganese toxicity (Prakash et al. Biorxiv 2023). However, the MIC of BCM for WT BW25113 is 37µg/ml (Malik et al. 2014), which lies approximately in the middle of 25 μg/ml and 50 μg/ml concentrations used in our study. This forms a good rationale for the concentrations used in present study. We have now justified the our chosen BCM concentrations in the manuscript in L4 of Results section. 

Results (L5): I disagree with this sentence: “We suspected that the reduced growth may be due to increased ROS levels in the cells after BCM treatment.” The main molecular mechanism of growth inhibition by BMC is already known (Rho inhibition). Please rewrite this sentence.

The sentence has been now modified.

I have important doubts with the quality of the images provided, at least in the version that I have been able to view. Please improve the quality of the images, especially Figure 1D, 4D, 5A and 4C. Additionally, I think that Figure 1F could be incorporated into Figure 2.

We have rechecked the image quality, all the uploaded images are of 300 dpi resolution. The images in manuscript draft that I downloaded also not looked as good quality that I submitted to the portal. In the figure 4.D the colocalization foci of DAPI and GFP has been now more precisely indicated.

An important aspect of the work is that it should be more clearly specified what type of reactive oxygen species are detected by the DHE and H2DCFDA probes. This should be described precisely in Methods. In relation to this, the Results statement "These results suggest that exposure to BCM also leads to the generation of highly deleterious hydroxyl radicals" is not clear. Why do we talk about "hydroxyl radicals" specifically if H2DCFDA is a generic ROS marker?

The ROS species detected by the two dyes are now described in the Materials and Methods. Thiourea is a ROS scavenger that is specific for hydroxyl radicals. So, when increased fluorescence of H2DCFDA after addition of BCM is quenched by supplementation of thiourea, it is safe to reason that BCM treatment leads to the generation of hydrioxyl radicals. 

Please delete this sentence from Discussion; “Although, we did not check the envelope stress after BCM treatment, it may be implicated in the generation of ROS after BCM treatment.” It is excessively speculative.

Earlier, it has been reported that misfolded proteins leads to ROS production in the cells through envelop stress response pathway (Kohanaski et al. 2008). Rho inhibition also results in unregulated protein expression. So, it seems reasonable to anticipate that Rho inhibition can potentially cause envelope stress. 

Methods: In relation to this sentence: “recAgfp was freshly transduced into the WT genome by P1 phage transduction to obtain the BW25113::recA-gfp strain (45).” Please, could you indicate the precise integration site in the bacterial chromosome?

When I looked at the reference article where construction of recA-gfp described, it turned out that the gfp coding sequence has been fused to recA promoter. So, when a P1 phage is transduced in BW25113, by homologous recombination the recA:gfp construct will recombine at the recA promoter in the bacterial chromosome.

It is necessary to include a Statistical Analysis section in the Methods section where the analyzes carried out are specified.

Statistical analyses is now presented in the Materials and Methods section

Minor points

Abstract: BMC should be used always after its definition

BCM is defined now in the Abstract as well as in L-6 in the Introduction text.

Abstract: “PAS” is not defined in any part of the text.

PAS is now defined in the Abstract section as well in L-5 of Discussion section.

Introduction:

L4: References (1-4) does not include information about “some researchers have negated the role of ROS…”

Information in the citations now has been included.

L6: BMC is mentioned at the fort time in the test, please define.

BCM is now defined there.

L8: Please change “gram-negative” to “Gram-negative”

“gram-negative” is now changed to “Gram-negative” 

Results:

L2: Please change “Effect of BCM” to “effect of BCM”

It has been changed.

Figure 1C: Please, change Soxs to SoxS

It has been changed.

Please change “TheBCM is known to cause the cell filamentation” to “The BCM is known to cause the cell filamentation”.

Iit has been changed.

Please change “upregulated many-fold in microarray after BCM treatment” to “upregulated up to XX-fold in microarray after BCM treatment”.

It has been changed.

Please change “in WT E. coli (Figure 5.B).. BCM” to “in WT E. coli (Figure 5.B). BCM”.

It has been changed.

Discussion

Please change “ampicillin, Norfloxacin, and vancomycin” to “ampicillin, norfloxacin, and vancomycin”.

It has been changed.

Please change “and H2DCFDA dye in Escherichia coli cells” to “and H2DCFDA dye in E. coli cells”.

It has been changed.

Please change “deciphered. Quinones, aminoglycosides” to “deciphered. Quinolones, aminoglycosides”.

It has been changed.

In the sentence: “MFI values obtained from three experiments”. MFI is not defined in any place.

Now, MFI has been defined under “Checking ROS level and induction of SOS response by flow cytometry” heading in the “Materials and Methods section”, where it appearead first.

In the sentence: “Forte Western HRP substrate (Millipore) was used to develop the”. Correct the end of the sentence.

It has been now corrected.

References

1.Cardinale CJ, Washburn RS, Tadigotla VR, Brown LM, Gottesman ME, Nudler E. Termination factor Rho and its cofactors NusA and NusG silence foreign DNA in E-coli. Science. 2008;320(5878):935-8.

2.Malik M, Li L, Zhao X, Kerns RJ, Berger JM, Drlica K. Lethal synergy involving bicyclomycin: an approach for reviving old antibiotics. J Antimicrob Chemother. 2014;69(12):3227-35.

3.Kohanski MA, Dwyer DJ, Wierzbowski J, Cottarel G, Collins JJ. Mistranslation of Membrane Proteins and Two-Component System Activation Trigger Antibiotic-Mediated Cell Death. Cell. 2008;135(4):679-90.

4.Prakash A, Kalita A, Kanika B, …Dutta D. Rho and a riboswitch regulate mntP expression evading manganese stress and membrane toxicity. Biorxiv. 2023

---

## [Decision Letter · Decision Letter 1]

27 Feb 2024

Bicyclomycin generates ROS and blocks cell division in Escherichia coli

PONE-D-23-34381R1

Dear Dr. Prakash,

We’re pleased to inform you that your manuscript has been judged scientifically suitable for publication and will be formally accepted for publication once it meets all outstanding technical requirements.

Kind regards,

Samiullah Khan, Ph. D

Academic Editor

PLOS ONE

Additional Editor Comments (optional):

Dear author,

The manuscript has been revised appropriately, both reviewrs are satisfied with the revised version. I recommend the manscript is now acceptable for publication in PLoS ONE.

Reviewers' comments:

Reviewer's Responses to Questions

**Comments to the Author**

1. If the authors have adequately addressed your comments raised in a previous round of review and you feel that this manuscript is now acceptable for publication, you may indicate that here to bypass the “Comments to the Author” section, enter your conflict of interest statement in the “Confidential to Editor” section, and submit your "Accept" recommendation.

Reviewer #1: (No Response)

Reviewer #2: All comments have been addressed

2. Is the manuscript technically sound, and do the data support the conclusions?

Reviewer #1: Partly

Reviewer #2: Yes

3. Has the statistical analysis been performed appropriately and rigorously? 

Reviewer #1: Yes

Reviewer #2: Yes

4. Have the authors made all data underlying the findings in their manuscript fully available?

Reviewer #1: Yes

Reviewer #2: Yes

5. Is the manuscript presented in an intelligible fashion and written in standard English?

Reviewer #1: Yes

Reviewer #2: Yes

6. Review Comments to the Author

Reviewer #1: The authors' main objective was to test whether the induction of oxidative stress could explain the increased sensitivity of Escherichia coli to bicyclomycin. It was shown that BCM indeed caused the formation of ROS, inducing cell filamentation as part of the SOS response. The results of the work allow us to develop more effective approaches to the use of BCM in clinical practice. The authors generally took into account my comments, however, considering the methods and approaches they used in this work, the question of the role of ROS in the action antibiotics remains controversial. According to the reviewer, in the future it is necessary to accumulate additional and more convincing evidence to solve the problem. The foregoing does not diminish the merits of the work under review.

Reviewer #2: (No Response)

7. PLOS authors have the option to publish the peer review history of their article (what does this mean?). If published, this will include your full peer review and any attached files.

Reviewer #1: No

Reviewer #2: **Yes: **José Manuel Rodríguez Martínez

---

## [Editor Report · Acceptance letter]

21 Mar 2024

PONE-D-23-34381R1 

PLOS ONE

Dear Dr. Prakash, 

I'm pleased to inform you that your manuscript has been deemed suitable for publication in PLOS ONE. Congratulations! Your manuscript is now being handed over to our production team.

Kind regards, 

on behalf of

Dr. Samiullah Khan 

Academic Editor

PLOS ONE